# Probiotics, Prebiotics, and Synbiotics Utilization in Crayfish Aquaculture and Factors Affecting Gut Microbiota

**DOI:** 10.3390/microorganisms11051232

**Published:** 2023-05-07

**Authors:** Maria V. Alvanou, Konstantinos Feidantsis, Alexandra Staikou, Apostolos P. Apostolidis, Basile Michaelidis, Ioannis A. Giantsis

**Affiliations:** 1Department of Animal Science, Faculty of Agricultural Sciences, University of Western Macedonia, 53100 Florina, Greece; mariaalvanou7@gmail.com; 2Laboratory of Animal Physiology, Department of Zoology, Faculty of Science, School of Biology, Aristotle University of Thessaloniki, 54124 Thessaloniki, Greece; kfeidant@bio.auth.gr (K.F.); michaeli@bio.auth.gr (B.M.); 3Laboratory of Marine and Terrestrial Animal Diversity, Department of Zoology, Faculty of Science, School of Biology, Aristotle University of Thessaloniki, 54124 Thessaloniki, Greece; astaikou@bio.auth.gr; 4Laboratory of Ichthyology & Fisheries, Department of Animal Production, Faculty of Agriculture, Forestry and Natural Environment, Aristotle University of Thessaloniki, 54124 Thessaloniki, Greece; apaposto@agro.auth.gr

**Keywords:** crayfish, decapods, dietary supplementation, health, microbiome, gut, microbiota

## Abstract

Aquaculture is affected by numerous factors that may cause various health threats that have to be controlled by the most environmentally friendly approaches. In this context, prebiotics, probiotics, and synbiotics are frequently incorporated into organisms’ feeding rations to ameliorate the health status of the host’s intestine, enhancing its functionality and physiological performance, and to confront increasing antimicrobial resistance. The first step in this direction is the understanding of the complex microbiome system of the organism in order to administer the optimal supplement, in the best concentration, and in the correct way. In the present review, pre-, pro-, and synbiotics as aquaculture additives, together with the factors affecting gut microbiome in crayfish, are discussed, combined with their future prospective outcomes. Probiotics constitute non-pathogenic bacteria, mainly focused on organisms’ energy production and efficient immune response; prebiotics constitute fiber indigestible by the host organism, which promote the preferred gastrointestinal tract microorganisms’ growth and activity towards the optimum balance between the gastrointestinal and immune system’s microbiota; whereas synbiotics constitute their combination as a blend. Among pro-, pre-, and synbiotics’ multiple benefits are boosted immunity, increased resistance towards pathogens, and overall welfare promotion. Furthermore, we reviewed the intestinal microbiota abundance and composition, which are found to be influenced by a plethora of factors, including the organism’s developmental stage, infection by pathogens, diet, environmental conditions, culture methods, and exposure to toxins. Intestinal microbial communities in crayfish exhibit high plasticity, with infections leading to reduced diversity and abundance. The addition of synbiotic supplementation seems to provide better results than probiotics and prebiotics separately; however, there are still conflicting results regarding the optimal concentration.

## 1. Introduction

The growing global demand for animal protein due to the world population’s rapid growth has led to the prevalence of aquaculture in fish and shellfish production (~50% of global production) [1]. The aquaculture industry provides consumers with animal products of high quality that possess an increased protein percentage while simultaneously contributing to global food safety, and its production is estimated to increase further in tandem with the increasing demand [2]. In total, during the last 70 years, the amount of fish production destined for human consumption doubled in comparison to 1960 [3]. Further, aquaculture production and human consumption per capita demonstrate a sharper increase in comparison with other protein sources of animal origin [4]. There is evidence that the aquaculture industry has overcome fisheries production as a source of seafood [5]. However, the increased production is, again, not enough to meet the expected global demand in the future. Thus, an extensive intensification of the aquaculture production processes is needed, both technologically and practically [6]. The intensification level is not homogeneous at a global scale; in some countries, the intensification patterns are currently more advanced than in others [7]. China is considered the country where the aquaculture sector took its first steps, and today, it has evolved into one of the fastest-growing sectors among the food supply network [8].

Although crayfish aquaculture is a lower-profile sector in comparison to other aquaculture products, it presents a very promising potential. Crustaceans are considered one of the foods with the fastest worldwide growth rates. Crayfish aquaculture apart from having a lower carbon footprint in comparison with other fish aquaculture practices, also contributes to the development of the regional economy [9]. Crustaceans, which in less than two decades tripled the world’s shrimp production output, are considered among the foods with the fastest worldwide growth rates, according to the Boston Consulting Group’s (BCG) most recent report [10]. Although crayfish aquaculture exhibits lower production quantities compared to shrimp aquaculture [11], its potential to be both economically and environmentally beneficial is considerable [9]. Specifically, China has reported rapid development in its crayfish farming sector, with outputs exceeding 1 million tons in 2018 [12]. China is currently the world’s top crayfish producer, followed by the United States, Turkey, and the European Union, according to a report by the Food and Agricultural Organization (FAO) of the United Nations. Approximately one million metric tons of crayfish are produced globally, with China accounting for more than 95% of the total production [1,3]. Crayfish meat is considered a delicacy in many countries due to its high protein quality and fatty acid profile [13]. The huge demand for crayfish in the international market, coupled with growing concerns about overfishing and the degradation of their natural habitats, has helped this sector to grow in prominence.

Crayfish has evolved into a profitable commodity in aquaculture [14,15], with a wide range of species brought up following a variety of technologically advanced approaches to harvest. These include white river crayfish *Procambarus zonangulus* (Hobbs & Hobbs, 1990), red swamp crayfish *Procambarus clarkii* (Girard, 1852), and other cambarids–*Cambarus robustus* (Girard, 1852), *Faxonius rusticus* (Girard, 1852), *Faxonius limosus* (Rafinesque, 1817), and *Faxonius virilis* (Hagen, 1870) [14,16]. Further, other cultured species in small operations are *Astacus astacus, Pontastacus leptodactylus*, and *Pacifastacus leniusculus*. Among the *Cherax* genus, four species are cultured: *Cherax destructor*, *Cherax quadricarinatus*, *Cherax albidus*, and *Cherax cainii* [17]. Overall, crayfish aquaculture presents a promising opportunity for economic development and food security while reducing pressure on natural crayfish populations.

Due to intensification and environmental deterioration, numerous diseases have emerged [18]. Many different substances, such as antibiotics, synthetic phenols, and insecticides, have been used to eliminate pathogens [19]. However, the overuse of the above has led to increased resistant pathogens [20]. Antibiotic-resistant bacteria in aquaculture represent a growing problem, and new strategies are needed to combat multidrug-resistant (MDR) bacteria [21]. The role of the microbiome in immune and neurologic development, growth, infections, and inflammatory diseases has been examined [22]. Intestinal microbiome disruption caused by plenty of external or dietary factors could lead to pathogen colonization [23]. As infections by MDR bacteria continue to grow as a major threat towards global health, gut microbiota poses a possible target for eliminating these threats [24]. Among the strategies applied in reared aquatic species, one of the most promising is the utilization of live microorganisms administered by injection, feed, or as water additives for controlling infectious diseases [25,26]. Particularly in decapods, prebiotics, probiotics, and synbiotics seem to play crucial roles, affecting various health and production factors. However, prior knowledge of microbial composition and interaction is needed in order to develop the most suitable supplement in each case. Previous review studies discuss the effect of these supplements in fish, shellfish, and shrimp aquaculture; however, a comprehensive review focusing on crayfish aquaculture is missing. Although there are occasionally contradictory results regarding the administration of pre-, pro-, and synbiotics in crayfish culture, there is no such comprehensive review combining all this information.

Therefore, the scope of the present study is to review the utilization of pre-, pro-, and synbiotics in crayfish aquaculture, their effects on crayfish farming, and proposed benefits and their mechanism of action, as well as to present some future perspectives. Furthermore, the microbiome of crayfish, apart from being a dynamic and complex biological system, exhibits a key role in many physiological processes of the organism [27]. Thus, in the present study, the factors affecting crayfish microbiota (including growth, diseases, and farming type) are reviewed and discussed.

## 2. Pro-, Pre-, and Synbiotics

Shellfish aquaculture’s intensification relies on the world’s strong appetite for shellfish; specifically, their rich valuable protein content and healthy profile of unsaturated fats [28], which, when substituting saturated fats (SFAs) and trans fats, decrease the risk of cardiovascular disease (CVD) [29,30]. However, the intensification of aquaculture practices exerts versatile stresses on cultured aquatic organisms [31], primarily due to infectious pathogens, which trigger their immune defense system responses and pose serious threats to the aquaculture industry’s growth and sustainability [32]. Since crustaceans’ productivity is highly dependent on their habitat, and therefore different environmental changes may cause viral and bacterial diseases, their stock’s conservation demands intensive and rigorous management [33]. Since sustainable cultivation is an integral and economically viable component of the aquaculture sector, shellfish aquaculture has largely relied on cutting-edge technologies, such as recirculating aquaculture systems (RAS), to effectively address threats posed by pathogens to shellfishes [34,35,36]. Additionally, aquaculture practices regarding the treatment of pathogens commonly and widely depend on antibiotics, which are administered in the diets of cultured aquatic species [32]. However, disease prevention treatments through antibiotics may increase opportunistic pathogens’ infestation [37] and favor antibiotic-resistant pathogens. The latter can pose serious negative health effects due to their ability to be transmitted toward terrestrial animals and enter the human food chain [38,39]. Moreover, indiscrete antibiotic application impedes the beneficial activity of gut microbiota. This can have serious effects on the physiological processes of cultured aquatic organisms, such as altered microbial systems, disrupted nutrition, and immunological competence [40,41].

Because antimicrobial agents’ health threats and adverse side effects have become uncomfortably apparent to both producers and consumers, alternative approaches are urgently needed to address the threat of pathogens in aquaculture [42,43]. Probiotics have been acknowledged as significant replacement agents for those stressors and for their detrimental effects, serving as immune modulators and boosting resistance to different microbial infections [42,43,44]. In this context, feed additives, such as prebiotics, probiotics, and synbiotics, are frequently incorporated into cultured aquatic organisms’ diets [5]. The aim of such practices is to considerably ameliorate the microbial and morphological health status of the host’s intestine [45,46], enhancing its functionality [47]. Prebiotics (indigestible fiber) increase the preferred gastrointestinal tract microorganisms’ growth and activity, and provide a clear balance between the gastrointestinal and immune system’s microbiota, thus benefiting the host’s immunity and health [48,49,50]. Probiotics (non-pathogenic bacteria-based products) are mainly focused on organisms’ energy production and efficient immune response [51], increasing resistance against various pathogens [42,43,44]. However, mounting evidence enhances their role in increased nutrients’ absorption, stress resistance, and fertility of the host species [52], thus shaping them as positive promoters of aquatic animal growth, survival, and health [53]. Lastly, the promotion of the gastrointestinal tract probiotics’ growth and survival by prebiotics in a synergistic “Synbiosis” relationship can provide an effective and rigorous management of the aquaculture sector [31,54]. Therefore, pro-, pre-, and synbiotics’ multiple benefits (strengthened immune responses, antibacterial agents’ growth, gut microflora alterations, competition for nutrients and binding sites, and enzyme-related activities) make these nutrients a valuable ally and thus a popular practice for the aquaculture industry [47].

Gram-positive probiotic bacteria are known for their effectiveness in controlling disease outbreaks in aquaculture. Gram-positive probiotics can lead to adverse effects on potentially pathogenic Gram-negative bacteria in the intestine of aquatic animals by secreting bioactive substances such as bacteriocins, siderophores, enzymes, and antibiotics. This creates a barrier against the attachment and colonization of disease agents in the gastrointestinal (GI) tract [55]. Gram-negative bacteria, including *Vibrio* and *Aeromonas,* are categorized among the most significant threats for disease outbreaks in aquaculture [56,57].

Compared to other finfish and shrimp species, few studies have investigated the efficacy and potency of lactic acid bacteria towards pathogenic Gram-negative bacteria in crayfish farming. Generally, *Lactobacillus* sp. bacteria isolated from goat milk are known for producing bacteriocins, which act as inhibitors against pathogens such as *V. harveyi*, *V. parahaemolyticus*, and *Aeromonas hydrophila* [58], which was also the case for crayfish [59,60].

Probiotics, including those among the *Bacillus* genus, have shown antagonistic activity towards a broad range of Gram-negative and Gram-positive bacteria. Their inhibitory effects can be attributed to many factors, such as the use of essential nutrients and changes in pH values, as well as the production of inhibitory substances (i.e., volatile compounds) [61]. Additionally, peptides produced by *Bacillus* sp. (bacitracin, polymyxin, gramicidin S, and tyrothricin) seem to have bioactive action against potential pathogens [55]. Thus, the evaluation of interactions of these supplements when they are administered in aquatic organisms, and more specifically in crayfish individuals, is of paramount importance.

## 3. Main Pre-, Pro-, and Synbiotics Substances Administered in Crayfish

Dietary, watery, or injected probiotic, prebiotic, and synbiotic supplements affect overall growth performance and susceptibility towards pathogens. Further, many studies highlight their action as immunomodulators, while boosting the immune system of the receiving organisms. Studies addressing the administrations of pro-, pre-, and synbiotic supplements in crayfish aquaculture are summarized in Table 1, Table 2 and Table 3, respectively.

### 3.1. Probiotics Administration

All information regarding the administration of probiotic supplements in crayfish aquaculture is summarized in Table 1. Singe probiotics such as *Bacillus subtilis* and *Bacillus licheniformis* exhibit positive effects on the immunity and survival rate of *P. leptodactylus* [79] and *P. clarkii* [74] when administered as dietary and water additives, respectively. After the administration of *Lactobacillus plantarum* on *P. leptodactylus* [63] and *C. cainii* [60,70], positive results in immunity parameters were observed in both, while in *C. cainii*, the diversity of intestinal microbiota increased. However, severe histopathological effects in both the guts and hepatopancreas were observed when non-industrial effective microorganisms were added to the diet of *P. leptodactylus*, while no effect was observed on their growth rate [76]. Furthermore, no positive effect on the growth and survival of stage II *P. leptodactylus* juveniles was observed when lactic acid bacteria and *Hafnia alvei* were applied both as dietary and water additives [77]. From another study on *P. clarkii*, a probiotic strain A23 *Bacillus amyloliquefaciens,* isolated from healthy individuals and added to the diet, demonstrated promising results, providing multiple benefits for crayfish cultivation. More specifically, it was found to enhance intestinal digestive enzyme activities, innate immune genes expression, and enzyme activities, as well as white-spot syndrome virus (WSSV) resistance [64,69]. Further, *B. amyloliquefaciens* supplements the decreased apoptosis of hemocytes [69]. The above results were further confirmed in a recent study investigating two other fish-derived probiotics, namely *Bacillus coagulans* (SCC-19) and *Lactococcus lactis* (Z-2), where increased activities of immune-related enzymes and mRNA expression of two AMP genes, better integrity, and a thicker mucosal layer, together with higher density granules in epithelial cells and increased phagocytosis rate of hemocytes and pathogen resistance, were observed [65]. Finally, intestinal microbiota diversity was found to be elevated [65]. Dietary *Limosilactobacillus fermentum* GR-3 revealed positive effects on *P. clarkii* gut microbiota, as it was observed that the dysbiosis incurred from Arsenic (As) reduced and further field application led to a significant increase in production [66]. In addition, dietary supplementation of *Saccharomyces cerevisiae* [73] resulted in increased weight gain, SGR, expression of LYZ, prophenoloxydase (proPO), and resistance towards *Citrobacter freundii*. Positive effects were also observed on the health status of *C. cainii* by the means of immune indices and microbial composition of the midgut after the dietary inclusion of *Bacillus* [67,80], *Clostridium butyricum* [78], and *Lactobacillus acidophilus* [60]. More specifically, *Holdemania* and *Vibrio* were identified as the most abundant bacteria in the groups fed the probiotic and in the control group, respectively [60], while *Lactobacillus* abundance was associated with the up-regulation of immune genes expression after the probiotic inclusion [70]. In *Cherax tenuimanus*, improved resistance towards *Vibrio mimicus* was also observed after dietary supplementation of probiotic bacteria (*Bacillus* sp.) (A10 (*Bacillus mycoides*), A12 (*Shewanella* sp.), PM3 (*B. subtilis*), and PM4 (*Bacillus* sp.)), whereas they were found to positively affect the physiological condition of crayfish with no impact on intermoult period, growth, and survival [68]. However, contradictory results were observed on *C. quadricarinatus,* where commercial probiotics [62], including *Bacillus, Acinetobacter*, and *Chryseobacterium* genera, could not control *A. hydrophila* in the system, while (Ecoterra^®^) [72] supplementation only led to an increase in some hemolymph parameters. When *Cambarellus montezumae was* studied, the dietary probiotic Spomune© inclusion resulted in increased survival and growth rate, as well as weight gain [71]. Furthermore, in the same species, *Lactobacillus* inclusion also resulted in increased final weight and improved overall welfare [75] (Table 1).

### 3.2. Prebiotics Administration

The administration of prebiotic supplements in crayfish aquaculture is summarized in Table 2. In *P. leptodactylus*, 75% dietary fishmeal substitution with *Chlorella vulgaris* showed the highest values of final weight, SGR, protein efficiency ratio (PER), protein productive value (PPV), in vivo apparent digestibility coefficients of organic matter (ADC_OM_), and in vivo apparent digestibility coefficients of crude protein (ADC_CP_), while the lowest FCR was observed. Additionally, with the *Chlorella* inclusion, activities of alkaline protease, lipase, amylase, PO, SOD, LYZ, and NOS were stimulated [82]. When Mannanoligosaccharide (MOS) and fructooligosaccharide (FOS) were added to the diet of narrow-clawed crayfish, a positive impact on crayfish immunological responses to air and bacterial exposure challenges, feed utilization, and growth performance was observed [83]. Additionally, in the same species, it was revealed that dietary galactooligosaccharide (GOS) exhibits advantageous effects on innate immunity, stress resistance, intestinal microbiota, and digestive enzyme activity, while no significant improvement in growth performance and survival was observed [84]. Concerning *P. clarkii*, dietary *Haematococcus pluvialis* administration was found to increase WGR, SGR, and hemolymph immune-related enzyme activities while leading to a malondialdehyde (MDA) content decrease [81]. Additionally, activities of alkaline protease, lipase, amylase, PO, SOD, LYZ, and NOS were promoted [81]. Further, *P. clarkii* fed with sulfated β-glucan revealed improved overall growth performance together with antioxidant capacity and immunity. Additionally, the intestinal flora improved as abundances of beneficial probiotics increased, while those of maleficent decreased [87]. On red claw crayfish *C. quadricarinatus, an* injection of 3-HB with a monomer of poly-β-hydroxybutyrate (PHB) caused improved phagocytosis, suppressed the growth of pathogenic bacteria, and increased the expression of microtubule-related genes. Hence, this prebiotic helped the crayfish individuals to be more resistant to pathogens [85] overall. Similarly, in the same genus, *C. tenuimanus* (Smith, 1912), the prebiotic MOS’ inclusion in the diet led to survival, health status, and immunity improvement, especially under certain circumstances such as bacterial infection and stress conditions incurred by exposure to NH_3_ and air [86]. Dietary supplementation with Bio-Mos©, which has *S. cerevisiae* as a main ingredient [88], resulted in better WGR and SGR while also exhibiting positive results on the health status, intestinal microbiota composition, immune parameters, and disease resistance of *Cherax distructor* individuals (Table 2).

### 3.3. Synbiotics Administration

All information regarding the administration of synbiotic supplements in crayfish aquaculture is summarized in Table 3. Experiments conducted on *P. leptodactylus* individuals sought to evaluate the effects of prebiotics (galactooligosaccharide (GOS, MOS, and xylooligosaccharide (XOS)), probiotics (*Enterococcus faecalis* and *Pediococcus acidilactici*), and synbiotics on different physiological markers. Results indicate that crayfish fed with the GOS+ *Enterococcus* [89] and XOS + *E. faecalis* [90] diet revealed the highest activities of PO, SOD, LYZ, alkaline phosphatase (ALK), and NOS. Furthermore, after implementing the aforementioned diets, the survival of *A. hydrophila* exposure had increased [89,90]. These results highlight that crayfish fed with synbiotic-enriched diets had a better effect than a single administration with probiotics and/or prebiotics [89]. The above is in line with a previous study [59], where synbiotics *Lactobacillus salivarius* and pectin (PE) inclusion in the diet exhibited better results on growth performance, immunocompetence, and disease resistance in comparison to the single inclusion of prebiotics and probiotics separately in the diet. Experimental diets containing Biogen as probiotics, *Allium sativum* (garlic), *Cynodon dactylon* as immunostimulant, and sodium alginate as prebiotics revealed improvement in the growth and immune response of *P. clarkii* juveniles [91]. *Lactobacillus* sp. dietary administration, together with coconut pulp, operating as a prebiotic for crayfish individuals, including in the *Cherax* genus, led to an increased growth rate but had no effect on survival [93] (Table 3). In addition, higher survival towards *V. mimicus* was observed in *C. cainii* fed poultry by-product meal, fermented by *Lactobacillus casei* and *S. cerevisiae*. From the same study, it was concluded that these dietary inclusions were beneficial to crayfish specifically related to microbial community and immune-related cytokines [92]. (Table 3).

### 3.4. Synopsis of Pro-, Pre-, and Synbiotics Administration and Limitations

Administration of probiotics, prebiotics, and synbiotics in crayfish aquaculture has increasingly gained attention. First, there is evidence that these supplements improve growth performance and feed utilization (Table 1, Table 2 and Table 3). Many parameters linked to growth and feed utilization have been examined (SGR, WG, FCR, PER, LER). These effects could be attributed to the provision of necessary nutrients and increased activity of digestive enzymes, which will further increase the digestibility of feed. More specifically, synbiotics referred to increased fat decomposition, which led to beneficial effects on growth parameters. Furthermore, in some cases, the intestinal morphology improved, leading to more efficient gut functions.

Apart from digestive enzymes, the above supplements were found to enhance antioxidant enzymes (CAT, SOD, GRx). These enzymes operate as barriers towards oxidative stress, reducing the harmful effects of reactive oxygen species (ROS) and protecting the host against susceptibility to pathogens. More specifically, SOD led to decomposition of reactive O_2_− to H_2_O_2_, while CAT turned H_2_O_2_ into O_2_ and H_2_O [94]. GRx is an enzyme that protects the integrity of the cells by catalyzing the reduction between reduced glutathione and H_2_O_2_ [95].

Further, the administered substances exhibit immunostimulatory effects, as they were found to increase LYS, PO, proPO, and NOS activity. Additionally, increases were observed in other physiological parameters that depict immune modulation, such as THC, TVC, TPP, LGC, and SGC. However, no significant effects or adverse effects were obtained. More specifically, a severe pathological finding in both the guts and hepatopancreas was observed, combined with reduced survival. Still, the existing knowledge is insufficient with core information still missing. Hence, the administration of these supplements is not a simple process. With the exception of increased cost, attention is needed for their optimal application, as contradictory information exists regarding the optimal doses. Furthermore, the injected substances are not very practical due to the number of cultured individuals and the elevated stress caused to them. Thus, further research is needed for the clarification of the optimal doses, substances, and method of administration. Following this direction, the development of the optimal supplements in order to avoid the administration of substances operating as a threat to public health (i.e., antibiotics) requires prior knowledge of the organism’s microbiome. In comparison to shrimp, the crayfish microbiome is less studied [96], so further investigation is needed due to its high plasticity, and many factors influence its abundance and composition. The knowledge and understanding of the crayfish microbiota complex system have the potential to provide solutions for crayfish aquaculture.

## 4. Main Factors Affecting Crayfish Microbiota Abundances and Composition

The intestinal microbiome of crayfish and of all aquatic organisms in general is a dynamic and complex biological system that plays a key role in physiological functions. Additionally, the microbiota of aquatic organisms are closely related to environmental factors, with water ranking among the most important [27]. Most studies investigating crayfish microbiota alterations and compositions have been conducted primarily on four species, *P. clarkii*, *C. quadricarinatus*, *C. cainii*, and *P. leniusculus*. The main results of these studies concerning the main phylum and genera abundances are summarized in Figure 1, Figure 2, Figure 3, Figure 4 and Figure 5. Considering the broad range of crayfish species in comparison with its wide distribution, a clear conclusion cannot be drawn so far. However, investigating microbiota alterations, diversity, and composition is the first step towards an enhanced understanding of the interactions between the host, environment, and microbes.

### 4.1. P. Clarkii

As *P. clarkii* represents one of the most extensively cultured crayfish species [118], many factors have been addressed in order to assess their influence on microbiota (Figure 1 and Figure 2). When *P. clarkii’s* intestine microbiome from ponds and from rice co-culture fields was studied, no significant differentiation was found between the different breeding models [97]. However, conflicting results obtained from other studies that examined the same culture methods revealed a significantly different relative abundance of bacterial and archaeal communities in the gut of red swamp crayfish [119]. The most dominant phyla were Proteobacteria, Actinobacteria, Tenericutes, Firmicutes, Bacteroidetes [98], Cyanobacteria, Chloroflexi, Acidobacteria, RsaHF231 and Nitrospirae [97]. In fungal and viral communities, no significant differences were observed [119]. Furthermore, the abundance of intestinal microbiota in autumn was found higher than in the summer in both culture methods [81]. When ditchless rice–crayfish co-culture was compared with traditional rice–crayfish culture, it was revealed that it has a superior bacterial system, which led to a lower abundance of pathogen colonization in the crayfish’s intestine [120]. Bacterial communities of the environment and from the intestinal microbiota of *P. clarkii* as a host have been proposed to interact with each other [121].

Further, the bacterial communities in the hepatopancreas of *P. clarkii* at different health statuses, including healthy, anorexic, moribund, and whitish muscle statuses, were investigated, and distinct differences were found in the structure, composition, and predicted function of the hepatopancreatic microbiota between the healthy and sick crayfish. More specifically, the LEfSe analysis revealed that the synbiotic bacterial species that were significantly enriched were *Proteus penneri*, *Citrobacter sensu stricto*, and *Lactococcus garvieae,* and the potential probiotics, such as *Weissella cibaria* and *Lactobacillus murinus* in the healthy crayfish in comparison to sick crayfish, while the opportunistic pathogens, including *C. freundii*, *Plesiomonas shigelloides*, *Citrobacter sensu stricto* 7, and Terrisporobacter, in the hepatopancreas of sick crayfish were significantly more enriched than those of healthy crayfish. In addition, compared with that of healthy crayfish, the hepatopancreas of moribund crayfish had significantly enriched bacterial genera, such as *Dubosiella, Candidatus, Bacilloplasma*, and *Phreatobacter*, whereas the hepatopancreas of crayfish with whitish muscle disease was observed with a significant enrichment of some opportunistic pathogens, including *Morganella morganii*, *Providencia alcalifaciens*, *Vagococcus fluvialis*, *Clostridium lundense*, and *Bacteroides* [122]. Furthermore, the intestinal microbiota of *P. clarkii* individuals at different health statuses after WSSV infection (healthy crayfish (HC), WSSV-infected active crayfish (IAC), and WSSV-infected diseased crayfish (IDC)) demonstrated that the relative abundances of certain phyla changed significantly in WSSV-infected crayfish, as indicated by a decrease in Tenericutes, Firmicutes and an increase in Proteobacteria and Bacteroidetes in WSSV-infected groups [99,123]. The IAC group exhibited the highest species diversity [123], while the overabundance of *Aeromonas* and *Citrobacter* and the decrease in *Acinetobacter* and *Kurthia* were associated with severe WSSV disease [99,123]. In addition, significant differences were indicated in the composition of the gut microbiome after infection with *C. freundii,* which pose a threat to crayfish farming and can also cause human infection through consumption [124].

A key factor that seems to enact a crucial role in the intestinal microbiota of *P. clarkii* is thermal stress. More specifically, increased abundance of Proteobacteria and decreased abundance of Bacteroidetes and Firmicutes was observed as the temperature elevated. However, some adaptive mechanisms were also observed as the abundance of phyla Bacteroidetes and Firmicutes, and pathogenic genera *Shewanella* and *Acinetobacter* gradually decreased while the abundance of beneficial Tenericutes and *Rhodobacter* gradually increased [125]. Furthermore, the effects of cadmium (Cd) at different concentrations were investigated, and from the results, it was indicated that Cd exposure could induce intestinal histological damage and affect intestinal microbiota composition and functions [126]. A possible solution to Cd increased concentration could be the inoculation of probiotic *B. subtilis,* which was found to mineralize Cd and attenuate Cd accumulation in crayfish [105]. Except for Cd polystyrene and polyethylene nanoplastic, nitrite, and sulfide, mercury and Hepatotoxin microcystion-LR exposure seem to also have an adverse effect on intestinal microbiota [66,100,101,106,107]. The relative abundance of lactic acid bacteria, *Citrobacter*, and other probiotics decreased, while the relative abundance of some intestinal pathogens and some other genera such as *Shewanella* and *Acinetobacter* increased [66,106].

Additionally, the diversity of gut microbiota was found to decline during development stages, while a specific pattern was associated with each stage [108,127]. Except for developmental stage, diet seemed to affect relative abundance in the intestine microbiota of crayfish. More specifically, the main phyla identified in groups fed pelleted feed and extruded feed were Proteobacteria, Tenericutes, and Firmicutes. The composition of Proteobacteria in the intestine of the pelleted feed group was significantly lower in comparison with the extruded feed group [102]. A relative abundance of Bacteroidetes was also found to be higher in *P. clarkii* gut microbiota when fed with fermented feed [127]. Additionally, environmental conditions and sampling site are suggested to shape carapace microbiota, while gut microbiotas seem to be more stable and associated with the factors linked to the host [109].

### 4.2. Cherax Genus

Among the *Cherax* genus, *C. quadricarinatus* and *C. cainii* are the most popular species involved in crayfish farming. Thus, many factors affecting their microbiota abundance and composition have been studied (Figure 3 and Figure 4). Suspended zeolite, which is known for toxic metals uptake and nitrogenous waste filtering, has been proposed to improve the gut microbial diversity, metabolic functions, and immune response of the organisms [128]. For the same species, the long-term effects of starvation on health indices influence the gut microbiota and innate T immune response, indicating a significant modulation on the microbiome as the bacterial abundance at both genus and species level in post-starved marron, while core microbiota was replaced by *Vibrio* [110]. Further, significant differences were found in the composition of the gut microbiome after infection with a new-emerging viral pathogen, namely the Decapod iridescent virus 1 (DIV1) [111]. Interestingly, the effects of nanoplastics on *C. quadricarinatus* led to significant changes in gut microbiota, including a decrease in abundance of Bacteroidetes, Actinobacteria, and Firmicutes [115]. Other necessary studied parameters in assessing their effect on the intestinal microbiota of crayfish are supplementation with trace elements (manganese, silica, and phosphorus and two different biological filters, i.e., Gravel, Bio-Ball). From the results, it was observed that trace element supplementation at higher levels led to a significant increase in abundance of phosphate-solubilizing bacteria [129], while biological filters demonstrated higher microbial diversity in the gut of *C. cainii* [112].

### 4.3. Other Genera

Generally, in rice–crayfish culture, enriched microbes in crayfish gut from distinct sets are observed, which include *Shewanella*, *Ferroplasma*, *Leishmania*, and *Siphoviridae* genera [130]. Further, in rice co-culture fields, beneficial bacterial taxa, including *Bacillus* sp., *Streptomyces* sp., *Lactobacillus* sp., *Prevotella* sp., *Rhodobacter* sp., *Bifidobacterium* sp., *Akkermansia* sp., and *Lactococcus* sp., have been identified, while opportunistic pathogens, (*Citrobacter* sp., *Aeromonas* sp.) have been observed [131]. In *P. leptodactylus* individuals fed with diets including polyphenols extracted from olive mill wastewaters (OMWW), any pathological changes in the midgut and hindgut were found by histological analysis. In crayfish fed on an OMWW-enriched diet, total intestinal microbiota decreased, except for anaerobes and yeasts [132]. In the gut of *P. leniusculus,* high heterogenicity of bacterial abundance and composition among individuals has been demonstrated, while no significant alterations in the microbiome were revealed [116] following their exposure to environmentally relevant concentrations of sulfamethoxazole. *P. leniusculus* represents one of the most successful crayfish invaders in Europe, and as the microbiome plays a crucial role in the overall fitness of the host, it may also affect or be affected by the invasion range. Exoskeletal, hepatopancreatic, and intestinal microbiota exhibited differentiation among invasion core and invasion front populations [117] (Figure 5).

### 4.4. Overview of Factors Affecting Microbial Composition and Diversity and Limitations Existing

It can be drawn from all the studies summarized in Figure 1, Figure 2, Figure 3, Figure 4 and Figure 5 that the microbiome represents a very complex and dynamic system, with plenty of factors and conditions affecting it. The crayfish microbiome is characterized by high plasticity, as there are no strict patterns of microbial abundance and composition. Investigation of the crayfish intestinal microbiome is of major importance, as it is the first step towards the development of optimal supplements in order to eliminate the administration of substances operating as a threat to global health. Freshwater crayfish species have suffered from mass reduction events in their natural habitats, mainly due to anthropogenic effects (i.e., degradation of the natural environment; translocation of invasive species). Thus, the understanding of microbiome composition and alteration will shed more light onto successful invasions in new habitats, and will help not only towards conservation issues but also in attaining information regarding increased survival rates. Finding new ‘host-associated probiotics’, namely bacteria that are originally isolated from the rearing water or the GI tract of the host to improve the growth and health of the host [133], may be more effective than probiotics from other origins, but this is a research direction that requires further investigation.

The main phyla existing under any circumstances are Proteobacteria, Firmicutes, and Tenericutes. More specifically, Proteobacteria have been extensively observed in aquatic organisms and environments. This phylum includes a broad range of bacteria exhibited in the gut, with some of them operating as opportunistic pathogens (i.e., *Vibrio*, *Pseudomonas*) and causing diseases to crayfish as well [134,135]. Firmicutes represent another major phylum identified in the crayfish microbiome. In general, this genus includes Gram-positive bacteria that are used as probiotic supplements (i.e., Bacillus, Clostridium, and Lactobacillus genera) in crayfish aquaculture [70,78]. The next common phylum was Tenericutes, which included bacteria that have been found in plenty of organisms, including plants, vertebrates, invertebrates, and water, and have been observed as crucial components in intestinal health maintenance [136].

## 5. Conclusions and Future Perspectives

Aquaculture represents the fastest-growing sector of primary production, offering high-quality animal protein products that meet the demand for nutrition and food security. At the same time, an urgent need for alternative supplementation has arisen as the microbial resistance leading to global health threats increases. Thus, finding substances to replace or reduce antibiotics use is of major importance. Previously, many reviews have addressed the use of probiotics, prebiotics, and synbiotics in various aquaculture species [4,33,55,137,138,139,140,141,142,143]. However, no such study focused on crayfish species so far. Notably, in crayfish aquaculture, many pathogenic and viral diseases have been cataloged during the last few years [144]. The utilization of antibiotics in an effort to eliminate these diseases has led to problems concerning the health of both the animal host and consumers. Additionally, considering the rising global demand for sustainable and healthy products, the use of pro-, pre-, and synbiotics is of high importance as natural dietary supplements. These supplements were revealed to act in many beneficial ways, including boosting the immune system, increasing the resistance against pathogens, and improving the growth performance and overall well-being of the organisms (Figure 6). However, there are still many blur points, such as the selection of the appropriate probiotic strain and prebiotic type, as well as the appropriate combination for an optimum synbiotic combination. Further, it remains to be seen if the level of increase in weight gain and other growth parameters following the administration of these supplements can cover the rising demand rates. Most studies investigating the synergistic action of prebiotics and probiotics concluded that synbiotics supplements had better results than probiotics and prebiotics supplements, separately. Apart from a few exceptions [62,76], all the studies indicated that the supplementation with pre-, pro-, and synbiotics in crayfish farming provided positive results. However, many questions still exist regarding the optimal dose of the supplement. Further, in some cases, there are still questions regarding their efficacy as neither positive nor negative influences were observed. In addition to these supplements, other alternative additives can be included, such as paraprobiotics, i.e., non-biological part probiotics, plant extracts, algae, and byproducts with prebiotic properties. Finally, a more detailed investigation into the mechanism behind the beneficial observations and how these supplements affect the crayfish gut microbiome is highly desired.

One of the main mechanisms of action of probiotics towards immunity is by stimulating phagocytosis [146,147] as was found to promote the up-regulation of many defensive parameters (PO, SOD, LYZ, and NOS). In parallel, synbiotics enhance the increase in beneficial bacterial strains in the mucus and by competing for adhesion sites, preventing the growth of pathogenic strains [148]. Additionally, synbiotics facilitate the production of cytotoxic substances (such as cytokines). As far as increased susceptibility towards pathogens is concerned, probiotics produce siderophore substances and antimicrobial agents (antibiotics, antimicrobial peptides) [149]. Further, probiotics eradicate pathogens from the infected GI tract through competitive exclusion for nutrients and adhesion sites [150]. Additionally, from the literature, it occurs that probiotics mainly improve the overall growth by up-regulating the digestive enzymes, improving both feed utilization and digestibility [151], while at the same time, they influence the alteration of beneficial intestinal bacteria, which control the secretion of important digestive enzymes, and as a result, nutrients become more easily available to the organisms [5].

The gut microbiome contributes substantially to the development and physiological performance of the host, including the prevention of pathogen growth, immune system modulation, nutrient absorption, metabolic pathways regulation, and vitamin production [152]. Thus, analysis of microbiota is essential in the development of a sustainable aquaculture protocol. It is well established that the gut-associated microbiota of crustaceans are essential for preserving animal health and homeostasis. Therefore, it is of crucial importance to assess the impact of a wide range of factors on these microbial communities, especially in aquatic organisms [116]. However, regarding the complexity and the dynamics of microbial communities in aquatic animals’ microbiomes, the investigation of the associated factors is not an easy task. Factors leading to alterations in the abundance and composition of microbiota include diet, culture methods, pathogen infections, developmental stage, and toxin exposure (Figure 7). However, further studies are needed in order to better understand the relationship between microbial species and organisms’ health biomarkers, which will enable the mitigation of many diseases. The available molecular tools, such as DNA sequencing and NGS technology, including amplicon and shot-gun approaches, led to microbial communities’ identification and shed more light on the investigation of microbiota alterations. Further, the part of the gut studied is not consistent, as in some studies, the midgut, the hindgut, or the complete gut were used, leading to conflicting results as there are different microbial communities in each gut section. Thus, the investigation of the crayfish microbiome, both at abundance and diversity levels, requires consistent standards regarding the tissue type and technical processes in order to produce reliable and comparable results.

## Figures and Tables

**Figure 1 microorganisms-11-01232-f001:**
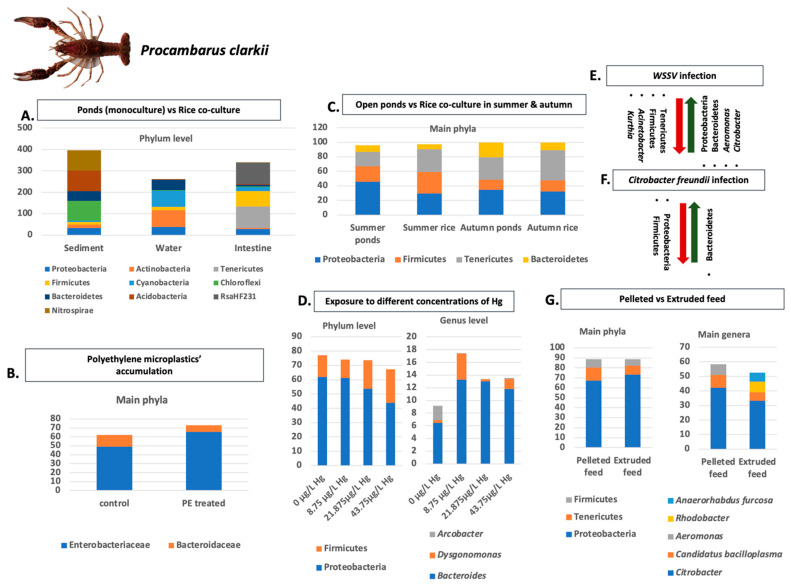
Main factors affecting *P. clarkii’s* microbiota: (**A**) Impact of Ponds Monoculture and Rice co-culture on microbiome of *P. clarkii* at phylum level; (**B**) Impact of polyethylene microplastics’ accumulations on main phyla of *P. clarkii*’s microbiome; (**C**) Differences in main phyla of *P. clarkii* microbiome when exposed to different culture types (open ponds and rice co-culture) and to different seasons; (**D**) Differences in main phyla and genera of *P. clarkii*’s microbiota after exposure to different Hg concentrations; (**E**) Abundance differences in main phyla and genera of *P. clarkii*’s microbiome after WSSV infection; (**F**) Abundance differences in main phyla of *P. clarkii*’s microbiome after infection with *Citrobacter freundii*; (**G**) Impact of pelleted and extruded feed on main phyla and genera of *P. clarkii*’s gut microbiome. Analyzed data obtained from Refs. [97,98,99,100,101,102,103]. *P. clarkii* photo, retrieved from [104].

**Figure 2 microorganisms-11-01232-f002:**
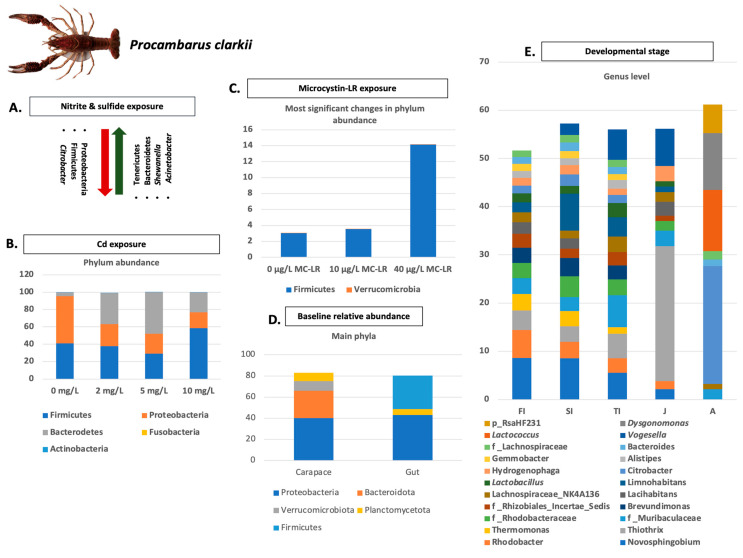
Main factors affecting *P. clarkii’s* microbiota (continued): (**A**) Impact of nitrite and sulfide exposure on abundance of main phyla and genera of microbiome of *P. clarkii*; (**B**) Impact of exposure to different Cd concentrations on main phyla abundance of *P. clarkii*’s microbiome; (**C**) Most significant changes of main phyla of *P. clarkii* microbiome when exposed to different concentration of microcystin-LR; (**D**) Baseline relative abundance in main phyla from two different *P. clarkii’s* tissues (carapace and gut); (**E**) Differences in abundance and composition in genera levels in P. clarkia individuals from different developmental stages. (FI): First instar larvae; (SI): Second instar larvae; (TI): Third instar larvae; (J): Juvenile; (A): Adult. Analyzed data obtained from Refs. [105,106,107,108,109]. *P. clarkii* photo retrieved from Ref. [104].

**Figure 3 microorganisms-11-01232-f003:**
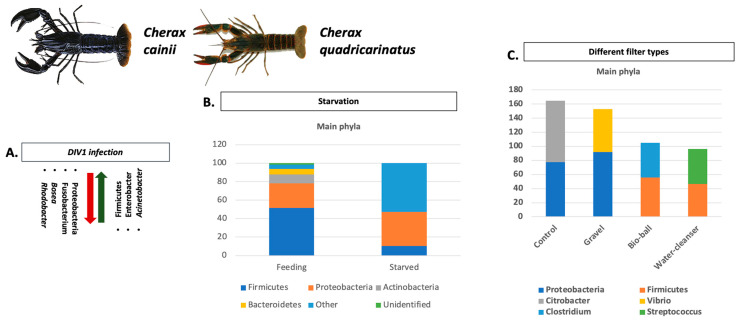
Main factors affecting microbiota of crayfish from *Cherax* genus: (**A**) Impact of DIV1 infection on abundance of main phyla and genera of microbiome of *C. quadricarinatus;* (**B**) Impact of starvation on main phyla abundance and composition of *C. cainii’s* microbiome; (**C**) Most significant changes of main phyla of *C. cainii gut* microbiome when cultured in water with different biological filters. Analyzed data obtained from Refs. [110,111,112]. *C. cainii* and *C. quadricarinatus* photos retrieved from Refs. [113,114], respectively.

**Figure 4 microorganisms-11-01232-f004:**
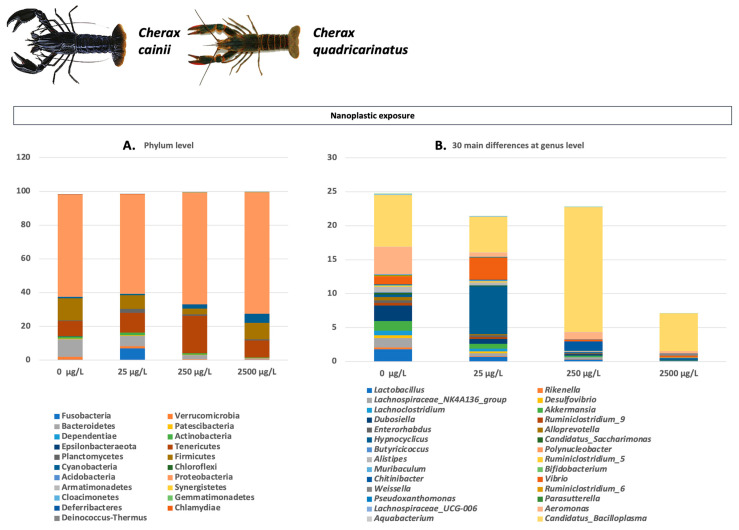
Main factors affecting microbiota of crayfish from *Cherax* genus (continued): (**A**) Differences in abundance and composition of gut microbiota in phylum level of *C. quadricarinatus* individuals after exposure to different nanoplastic concentrations; (**B**) 30 main differences at genus level of gut microbiota of *C. quadricarinatus* individuals after exposure to different nanoplastic concentrations. Data for Figures A and B obtained from Ref. [115]. *C. cainii* and *C. quadricarinatus* photos retrieved from Refs. [113,114], respectively.

**Figure 5 microorganisms-11-01232-f005:**
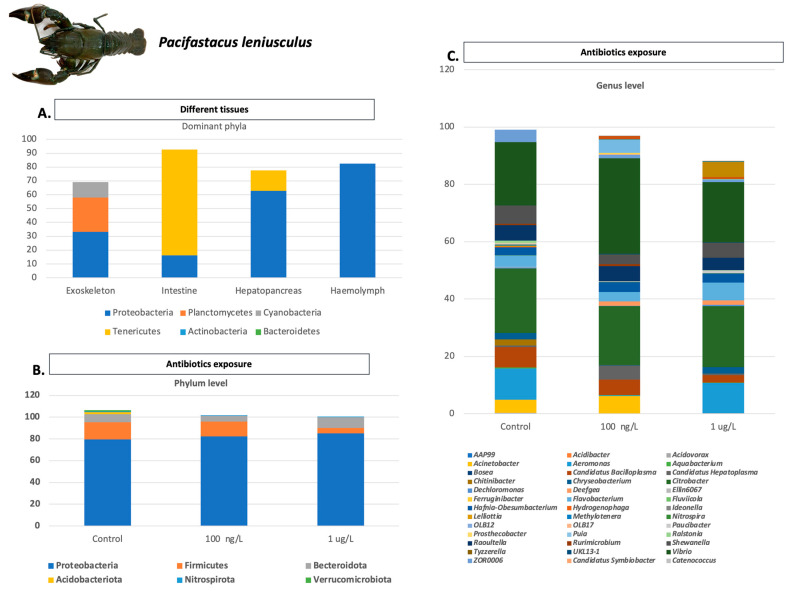
Main factors affecting microbiota of *P. leniusculus* individuals: (**A**) Differences in abundance and composition of microbiota in main phyla of *P. leniusculus* individuals in four different tissues (exoskeleton, intestine, hepatopancreas, and hemolymph); (**B**) Differences in abundance and composition of gut microbiota in phylum level of *P. leniusculus* individuals after exposure to different antibiotics concentrations; (**C**) Differences in abundance and composition of gut microbiota in genus level of *P. leniusculus* individuals after exposure to different antibiotics concentrations. Analyzed data obtained from Refs. [116,117]. *P. leniusculus* personal photo from Greece.

**Figure 6 microorganisms-11-01232-f006:**
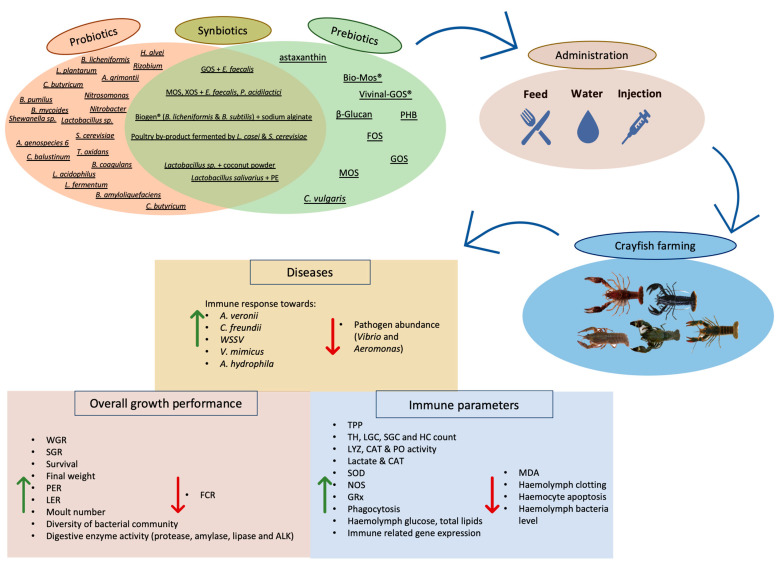
Effect of prebiotics, probiotics, and synbiotics on cultured crayfish. Many probiotics, including genera *Hafnia*, *Bacillus*, *Swawanella, Clostridium, Acinetobacter*, etc., and probiotics, including MOS, GOS, FOS, XOS, PHB, PE, astaxanthin, β-glycan, etc., as well as combination of the aforementioned two as synbiotics were applied in three main ways (through feeding, directly into the culture water or with injection). The administration of these substances resulted in overall improved growth performance and digestion indices. Further, immune parameters and resistance towards some common crayfish pathogens also improved after administration of prebiotics, probiotics, and synbiotics. *P. clarkii; C. quadricarinatus; C. cainii; P. leptodactylus;* photos retrieved from Refs. [104,113,114,145], respectively. *P. leniusculus* personal photo.

**Figure 7 microorganisms-11-01232-f007:**
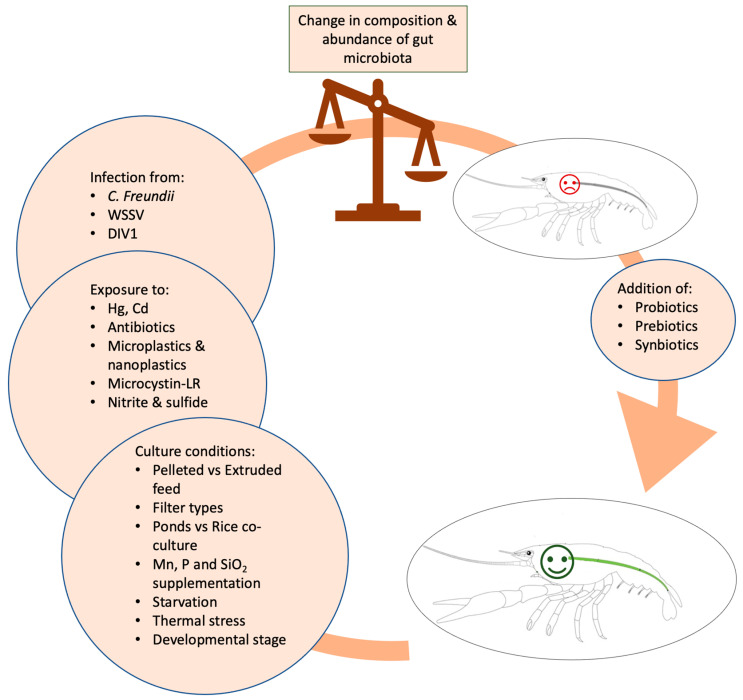
Main factors affecting abundance and composition of crayfish’s gut microbiome.

**Table 1 microorganisms-11-01232-t001:** Probiotics administration for evaluating their action towards crayfish aquaculture.

Probiotic	Source	Species	Administration	Concentration and Duration	Responses	Refs.
*Bacillus pumilus; B. licheniformis; B. subtilis; Acinetobacter genospecies 6, Acinetobacter grimontii* and *Chryseobacterium balustinum*	Commercial Bacillus probiotics, and other three from red clawed larvae	*Cherax quadricarinatus*	Inhibition test	104 CFU/mL	∙ unable to outgrow and out-compete pathogenic bacteria *A. hydrophila*	[62]
*L. plantarum*	Intestine of rainbow trout	*Astacus leptodactylus*	Dietary	107 (LB7), 108 (LB8), and 109 (LB9) CFUg^−1^ for 97 days	LB7 and LB8: ↑ TPP and PO LB8: ↑ LGC count ↑ LYZ activity LB7: ↑ SOD All LB groups: ↑ autochthonous LAB levels, lipase activity ↑ THC, SGC, and HC count LB8 and LB9: ↑digestive enzyme activity (protease, amylase, and ALP) LB7 and LB9: ↑ catalase (CAT) activity ∙ no significant growth ∙ no mortality after 24 h air exposure	[63]
A23 (*B. amyloliquefaciens*)	Intestine of healthy *P. clarkii*	*Procambarus clarkii*	Dietary	1 × 107 CFU/g (A23–1) and 1 × 108 CFU/g (A23–2) for 28 days	↑ intestinal digestive enzyme activities, innate immune enzyme activities ↑ white-spot syndrome virus (WSSV) resistance ↓ the abundance of Proteobacteria with 108 CFU/g supplement ↑ the abundance of Firmicutes and Bacteroides ↑ intestinal microbial diversity	[64]
*B. coagulans* (SCC-19) and *L. lactis* (Z-2)	SCC-19 from the gut of common carp and Z-2 from the gut of *Cyprinus carpio L.*	*Procambarus clarkii*	Dietary	106, 107, and 108 CFU/g for 28 days	↑ activities of immune-related enzymes in intestine ↑ mRNA expression of two AMP genes in intestinal ↑ intestinal integrity, thicker mucosal layer ↑ density granules in epithelial cells ↑ diversity of intestinal microbiota ↑ phagocytosis rate of hemocytes and pathogen *A. hydrophila* resistance	[65]
*L. fermentum* GR-3	Chinese traditional fermented food (Jiangshui noodles)	*Procambarus clarkii*	Dietary	1 × 108 CFU/g for 30 days	↓As(III) concentration and residual level in hepatopancreas by 36% ↓ gut microbiota dysbiosis due to As(III) exposure ↓ As(III) accumulation in field ↑ aquaculture production	[66]
*B. mycoides*	Provided by Department of Agriculture and Food, Western Australia	*Cherax cainii*	Dietary	108 CFU/g for 60 days	↑ health and immune indices (protein and energy in tail muscle, THC in hemolymph) ↑ different microbial communities in hindgut ↑ cytokine genes expression associated with immunity and health status	[67]
AQ2 (*Bacillus* sp.); A10 (*B. mycoides*); A12 (*Shewanella* sp.); PM3 (*B. subtilis*); and PM4 (*Bacillus* sp.)	A10 and A12 from healthy farmed marron intestines; AQ2 from Aquasonic Pty. Ltd. New South Wales, Australia; PM3 and PM4 from Enviroplus Pty Ltd. Singapore	*Cherax tenuinamus*	Dietary	108 CFU/g for 70 days	∙ no significant impact on survival, growth, and intermoult period ∙ physiological condition in tail muscle indices, proportion of GC, and THC ∙ bacteria in hemolymph ∙ bacterial community in gut ∙ resistance towards *V. mimicus*	[68]
*B. amyloliquefaciens*	Zhejiang Science and Technology University, Zhejiang, China	*Procambarus clarkii*	Dietary	4, 5, 6 g/kg for 7 days	∙ immune-related genes expression ∙ immune parameters (THC, PO activity, and SOD activity) ∙ hemocyte apoptosis ∙ innate immunity regulation ∙ mortality rate towards challenge with WSSV	[69]
*L. plantarum*	Quest L. plantarum, Nutra Pharma, West Yorkshire, UK	*Cherax cainii*	Dietary	1 × 109 CFU mL^−1^/kg for 56 days	∙ hemolymph parameters and gut health ∙ expression of innate immune response genes ∙ diversity of gut microbiota	[70]
Spomune© (*B. subtilis* and *C. butyricum*)	Not mentioned	*Cambarellus montezumae*	Dietary	1 × 107 CFU/g for 24 weeks	∙ survival, growth, and weight gain	[71]
(Ecoterra^®^) composed of *B. licheniformis*, *B. subtilis*, *Nitrobacter*, *Nitrosomonas*, *Rizobium*, *Saccharomyces cereviciae*, and T. oxidans	Not mentioned	*Cherax quadricarinatus*	Water additive	200,000 cells/liter for 60 days	∙ no effect on growth, FCR, and survival ∙ mean value of total lipids in hemolymph, hemolymph glucose, and total lactate	[72]
*S. cerevisiae*	Intestinal tract of crayfish	*Procambarus clarkii*	Dietary	107 CFU/g for 28 days	∙ weight gain, SGR, expression of lysozyme and prophenolxidase ∙ abundance of *Cetobacterium* and *Lactobacillus* ∙ abundance of *Citrobacter* and *Bacteroides* ∙ resistance towards *C. freundii*	[73]
*B. subtilis* CK3	Intestine of *P. clarkii*	*Procambarus clarkii*	Water additive	1 × 105 CFU/mL for 4 weeks	∙ antioxidant and immune-related enzymes and enzymes activities in hepatopancreas ∙ mortality ∙ immune response of *P. clarkii* towards *A. veronii*	[74]
*Lactobacillus* sp.	Digestive tract of angel fish *Pterophyllum scalare*	*Cambarellus moctezumae*	Dietary	100 mL of *Lactobacillus* solution (La3) for 24 weeks	∙ overall well-being ∙ final weight	[75]
Effective microorganisms’ serum with two major microorganisms as *B. amyloliquefaciens* spp. and *L. plantarum*	Rice-washed water	*Astacus leptodactylus*	Dietary powder and water additive	1% and 5% of powder in diet and 0.01% serum in water for 60 days	∙ no significant difference in growth performance ∙ severe pathological finding in both guts and hepatopancreas (inflammatory cell infiltrations in interstitial tissue, and lack of B, F, and R epithelial cells) ∙ survival rate	[76]
*H. alvei*	Hepatopancreas, gills, and intestine of adult crayfish and whole body of stage II crayfish juveniles and rearing water of adult and juvenile crayfish	*Astacus leptodactylus*	Diet and water additive	lactic acid bacteria (0.015 gL^−1^); H. alvei (106 CFU mL^−1^) and H. alvei added to water (106 CFU mL^−1^)	∙ no significant impact on growth and survival	[77]
*C. butyricum*	Advanced Orthomolecular Research (AOR, Calgary, AB, Canada)	*Cherax cainii*	Dietary	107 CFU/mL per kg for 42 days	∙ moult number, growth rate, THC, LYZ activity in hemolymph and protein content of tail muscle ∙ diversity of bacterial community ∙ Clostridium abundance ∙ crayfish pathogen abundance (*Vibrio* and *Aeromonas)* ∙ expression level of immune-responsive gene towards challenge with *V. mimicus*	[78]
Subtilis-C (*B. subtilis, B. licheniformis*)	Not mentioned	*Pontastacus leptodactylus*	Dietary	1.5 g per 1 kg of feed	∙ immunity, survival rate ∙ ACC of lysosomal cationic protein in hemocytes	[79]
*L. acidophilus* and *L. plantarum*	Nature Way Probiotic (Warriewood, New South Wales, Australia)	*Cherax cainii*	Dietary	109 CFU/mL per kg for 60 days	∙ no significant differences in weight gain ↑ hemolymph parameters and biochemical composition of tail muscle, hepatopancreas health ↑ microvilli counts ↑ shift of beneficial microbial communities ↑ metabolic functions and genes associated with innate immune response	[60]
*B. mycoides*	Marron origin	*Cherax cainii*	Dietary	108 CFU/g of feed for 10 weeks	↑ survival at 48 h of transport ↑ intestinal bacterial population and THC ↑ hemolymph bacteria (bacteraemia) level	[80]

Abbreviations: total plasma protein (TPP), phenoloxidase activity (PO), large granular cells (LGC), lysozyme (LYZ), superoxide dismutase (SOD), lactic acid bacteria (LAB), semi-granular cells (SGC), granular cells (GC), total hyaline cells (THC), hyaline cells (HC), alkaline phosphatase (ALP), catalase (CAT), average cytochemical coefficient (ACC), specific growth rate (SGR), feed conversion Ratio (FCR), white-spot syndrome virus (WSSV).

**Table 2 microorganisms-11-01232-t002:** Prebiotics administration for evaluating their action towards crayfish aquaculture.

Prebiotic	Source	Species	Administration	Concentration and Duration	Responses	Refs.
Astaxanthin	*H. pluvialis*	*Procambarus clarkii*	Dietary	o.6%	↑ WGR, SGR, and haemolymph immune-related enzyme activities ↑ MDA ↑ microbial dysbiosis and gut immune damage	[81]
*Chlorella vulgaris*	*Chlorella vulgaris*	*Pontastacus leptodactylus*	Dietary	75% substitution for 63 days	↑ final weight, SGR, PER, ADC_OM_ and ADC_CP_ ↑ alkaline protease, lipase, amylase, PO, SOD, LYZ, and NOS activity ↑ FCR dietary fishmeal substitution level (%) for maximum growth, SGR, and weight gain values	[82]
MOS and FOS	MOS, immunogen^®^, International Commerce Corporation Co., Waltham, MA, USA and FOS, Raftilose^®^ P95, Orafti Co., Tienen, Belgium	*Pontastacus leptodactylus*	Dietary	1.5, 3.0 and 4.5 g kg^−1^ in the single diets and 0.75, 1.5 and 2.25 g kg^−1^ in the combined diets for 126 days	↑ SGR, VFI, survival rate and ↑ FCR values in 2.25 g kg^−1^ MOS and 1.5 g kg^−1^ FOS ↑ PER, LER, EER, PPV, LPV and EPV in 2.25 g kg^−1^ MOS and 1.5 g kg^−1^ FOS ↑ amylase, lipase, and alkaline protease activities and the mean of hemolymph indices in 2.25 g kg^−1^ MOS and 1.5 g kg^−1^ FOS ↑ activities of PO, SOD, LYZ, and NOS after 12-h air exposure challenge in combined diets	[83]
Prebiotic Vivinal-GOS^®^ (rich in GOS)	Friesland Foods Domo Company (Zwolle, The Netherlands)	*Pontastacus leptodactylus*	Dietary	0, 1, 2, and 3% GOS for 97 days	↑ THC, SGC, and HC counts in 2% GOS diets ↑ CAT and CAT activity in 3% GOS diet ↑ LYZ, amylase and lipase activity, LAB levels, in 2% and 3% GOS-enriched diets ↑ THC, SGC, and HC count in 1% and 2% GOS diets ↑ total intestinal heterotrophic bacteria (TIHB) in the first 14 days in all GOS diets	[84]
PHB monomer (3-HB)	(166,898, Sigma Darmstadt, Germany)	*Cherax quadricarinatus*	Injection	5 × 10 CFU/mL	↑ phagocytosis, expression of microtubule-related genes ↑ growth of pathogenic bacteria	[85]
MOS (Bio-Mos^®^)	cell wall of *S. cerevisiae*	*Cherax tenuinamus*	Dietary	0.2% and 0.4% Bio-Mos^®^ for 30 days, 112 days for *V. mimicus* challenge, and 0.4% for 42 days for NH_3_ challenge	↑ survival after bacterial infection and exposure to NH_3_ ↑ unaltered THC after bacterial infection ↑ THCs after exposure to NH_3_ ↑ unaltered *Vibrio* spp. in hemolymph after bacterial infection and exposure to NH_3_ ↑ Hemolymph clotting time in Bio-Mos^®^ diet	[86]
β-Glucan	*S. cerevisiae*	*Procambarus clarkii*	Dietary	0.025%, 0.05%, 0.1%, and 0.2% for 8 weeks	↑ growth performance, antioxidant capacity, immunity, function and structure of the intestinal flora ↑ probiotics abundances of *Hafnia, Acinetobacter* ↑ probiotics abundance of *Enterobacteriaceae* ↑ *Aeromonas* abundance	[87]
MOS (Bio-Mos^®^)	Alltech	*Cherax destructor*	Dietary	0.4% for 56 days	↑ weight, SGR, and average weekly gain ↑ THC, GC, and SGC growth parameters ↑ protease activity in hepatopancreas ↑ amylase activity in the guts	[88]

Abbreviations: phenoloxidase (PO), lysozyme (LYZ), superoxide dismutase (SOD), lactic acid bacteria (LAB), semi-granular cells (SGC), granular cells (GC), total hyaline cells (THC), hyaline cells (HC), catalase (CAT), voluntary feed intake (VFI), nitric oxide synthase (NOS), malondialdehyde (MDA), galactooligosaccharide (GOS), mannanoligosaccharide (MOS), fructooligosaccharide (FOS), lipid efficiency ratio (LER), protein efficiency ratio (PER), specific growth rate (SGR), feed conversion ratio (FCR), energy efficiency ratio (EER), protein productive value (PPV), lipid productive value (LPV), energy productive value (EPV), apparent digestibility coefficients of organic matter (ADC_OM_), apparent digestibility coefficients of crude protein (ADC_CP_), poly-β-hydroxybutyrate (PHB), white-spot syndrome virus (WSSV).

**Table 3 microorganisms-11-01232-t003:** Synbiotics administration for evaluating their action towards crayfish aquaculture.

Synbiotic	Source	Species	Administration	Concentration and Duration	Responses	Refs.
GOS+ *Enterococcus faecalis*	*Enterococcus faecalis* from gastrointestinal tract of aquatic species GOS from dairy products	*Astacus leptodactylus*	Dietary	7.53 log CFU *E. faecalis* g^−1^ + 10 g kg^−1^ GOS for 126 days	↑ SGR, VFI, survival rate ↑ FCR ↑ in vivo apparent digestibility coefficients ↑ ratios of presumptive autochthonous LAB to total viable aerobic heterotrophic bacteria ↑ PO, SOD, LYZ, and NOS activity ↑ mean survival rate towards *A. hydrophila*	[89]
prebiotics (MOS and XOS); probiotics (*E. faecalis* and *P. acidilactici*) and synbiotics	MOS from International Commerce Corporation Co., USA; XOS from Shandong Longlive Bio-Technology Co., China; *E. faecalis* from Nichi Nichi Pharmaceutical Co., Ltd., Japan; *P. acidilactici* (Bactocell^®^, Lallemand Inc., Montreal, QC, Canada)	*Astacus leptodactylus*	Dietary	10 g kg^−1^ for prebiotics and 7.86 log CFU g^−1^ for probiotics for 126 days	XOS + *E. faecalis*: ↑ antibacterial activities in the shell mucus against *Nocardia brasilience* ↑ protein levels ↑ ALP and LYZ activities ↑ resistance after *A. hydrophila* injection ↑ growth rate and resistance to the *A. hydrophila* injection *MOS + P. acidilactici*: ↑ antibacterial activities in the shell mucus against *Vibrio harveyi* Both synbiotic diets: ↑ ratio of the *Lactobacillus* count to the total viable count	[90]
Biogen^®^ (*B. licheniformis* and *B. subtilis*) + sodium alginate	cell walls of brown seaweed	*Procambarus clarkii*	Dietary	1%, 2%, 3% Biogen^®^ and (3 g/L) of sodium alginate	↑ survival, wet weight, SGR, hemocyte count. and proPO activity	[91]
* L. salivarius * (LS) ATCC 11741 + PE	PE from Sigma-Aldrich Inc; *L. salivarius* from the Iranian Biological Resource Center	*Postantacus leptodactylus*	Dietary	LS1 (1 × 10^7^ CFU/g), LS2 (1 × 10^9^ CFU/g), PE1 (5 g/kg), PE2 (10 g/kg), LS1PE1 (1 × 10^7^ CFU/g + 5 g/kg); LS2PE2 (1 × 10^9^ CFU/g +10 g/kg) for 18 weeks	In all diets: ↑ final weight, weight gain, SGR and FCR ↑ TVC and LAB ↑ resistance towards *A. hydrophila* LS1PE1 and LS2PE2: ↑ amylase and protease enzymes activity ↑ GPx and SOD activity ↑ MDA content LS1PE1: ↑ THC, LGC, SGC, and HC count ↑ LYZ, PO, NOS, and AKP activity	[59]
Poultry by-product fermented by *L. casei* and *S.cerevisiae*	Poultry by-product from Specialty Feeds Pty. Ltd., Western Australia; *L. casei* and *S. cerevisiae* from Baker’s yeast	*Cherax cainii*	Dietary	75% substitution for 70 days	↑ no significant difference in final weight ↑ intestinal microvilli number ↑ *Lactobacillus* and *Streptococcus* in the intestine ↑ *Aeromonas* number in the intestine ↑ cytokines expression ↑ LYZ and phagocytic activity ↑ survival towards challenge with *V. mimicus*	[92]
*Lactobacillus* sp. and coconut powder	Coconut powder from coconut pulp from agricultural wastes; *Lactobacillus* sp. not mentioned	*Cherax* sp.	Dietary	*Lactobacillus* 2%/kg of feed + coconut powder 2%/kg feed	↑ growth rate ↑ no significant differences in survival rate	[93]

Abbreviations: prophenoloxidase (proPO), phenoloxidase (PO), large granular cells (LGC), lysozyme (LYZ), superoxide dismutase (SOD), lactic acid bacteria (LAB), semi-granular cells (SGC), total hyaline cells (THC), hyaline cells (HC), alkaline phosphatase (ALP), catalase (CAT), voluntary feed intake (VFI), nitric oxide synthase (NOS), total heterotrophic bacteria count (TVC), glutathione peroxidase (GPx), malondialdehyde (MDA), galactooligosaccharide (GOS), mannanoligosaccharide (MOS), xylooligosaccharide (XOS), fructooligosaccharide (FOS), pectin (PE), specific growth rate (SGR), feed conversion ratio (FCR).

## Data Availability

No new data were created or analyzed in this study.

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
