# Peer review of "Probiotics, Prebiotics, and Synbiotics Utilization in Crayfish Aquaculture and Factors Affecting Gut Microbiota"

_microorganisms, 2023, doi:10.3390/microorganisms11051232_

Round 1

Reviewer 1 Report

The Review entitled "Probiotics, prebiotics and synbiotics utilization in crayfish aquaculture and factors affecting gut microbiota" features as a review the utilization of pre- pro- and syn-biotics in crayfish culture, their categorization, proposed benefits, and perspectives. As the authors describe, the literature reports many information’s on the Pro-, pre-, and synbiotics’ multiple benefits, i.e. boost immunity, increase resistance towards pathogens, gut microflora alterations and overall welfare promotion, make these nutrients a valuable ally and thus a popular practice for the aquaculture industry. The review, however, a little must be improved in terms of writing since some grammar errors are present in the manuscript. The manuscript needs restructuring and substantially revised.

The authors should critically discuss the existing literature, point out the knowledge gaps, and suggest further research. The manuscript slightly lacks coherence in storyline, and English language also needs careful editing for better readability.

The manuscript needs restructuring and substantially revised.

Author Response

The Review entitled "Probiotics, prebiotics and synbiotics utilization in crayfish aquaculture and factors affecting gut microbiota" features as a review the utilization of pre- pro- and syn-biotics in crayfish culture, their categorization, proposed benefits, and perspectives. As the authors describe, the literature reports many information’s on the Pro-, pre-, and synbiotics’ multiple benefits, i.e. boost immunity, increase resistance towards pathogens, gut microflora alterations and overall welfare promotion, make these nutrients a valuable ally and thus a popular practice for the aquaculture industry. The review, however, a little must be improved in terms of writing since some grammar errors are present in the manuscript. The manuscript needs restructuring and substantially revised.

Response: In accordance to suggestion of the 1st reviewer, the manuscript was been totally restructured. The first two chapters were embedded in one, and the focus was moved towards crayfish aquaculture while all the general information was deleted. Secondly, in the chapter “Main pre-, pro- and synbiotics substances administered in crayfish” the information was completely restructured. More specifically, whereas previously the information was divided into sub-chapters according to the crayfish genera, in the revised manuscript the information has been separated into sub-chapters based on the administered substance type (please see sections 3.1-3.4). Further, the tables in the revised manuscript can be easier connected to a subchapter depending on whether they concern probiotics, prebiotics or symbiotics. In addition, two new paragraphs have been added at the end of the sections 3. "Main pre-, pro- and synbiotics substances administered in crayfish” and 4. "Main factors affecting crayfish microbiota abundancies and composition” in which the information previously mentioned in each chapter is summarized and critically discussed.

The authors should critically discuss the existing literature, point out the knowledge gaps, and suggest further research. The manuscript slightly lacks coherence in storyline, and English language also needs careful editing for better readability.

Response: According to the reviewer’s recommendation, the manuscript has been linguistically edited by a native speaker (attached an editing certification). Further two new subchapters were added (3.4 & 4.4) critically discussing all the aforementioned information, whereas in parallel, knowledge gaps are presented as well. Further, in the conclusion, parts discussing the existing literature, while in the same time pointing out the knowledge gaps and direction for future research were added.

Reviewer 2 Report

It is a comprehensive review discussing the probiotics, prebiotics and synbiotics utilization in crayfish aquaculture and factors affecting gut microbiota. The authors discussed crayfish aquaculture, composition of probiotics, prebiotics and synbiotics, main pre-, pro- and synbiotics substances administered in crayfish, and main factors affecting crayfish microbiota abundancies and composition.

Although the idea was previously published (PMID: 33971259, PMID: 34274422,...etc), the review is comprehensive

Major points:

1- The authors need to highlight  the new directions in this review that not published previously (i.e what are new data in this review that differs from previous published reviews discussing the same topic)

2-The authors mentioned in the Data Availability Statement: No new data were created or analyzed in this study . However, I see in the review data generated in Fig 1a, FIg 1b, Fig 2a, Fig 2b, and Figure. What are the source of these data used in these figures,

Moderate language editing is required

Author Response

It is a comprehensive review discussing the probiotics, prebiotics and synbiotics utilization in crayfish aquaculture and factors affecting gut microbiota. The authors discussed crayfish aquaculture, composition of probiotics, prebiotics and synbiotics, main pre-, pro- and synbiotics substances administered in crayfish, and main factors affecting crayfish microbiota abundancies and composition.

Although the idea was previously published (PMID: 33971259, PMID: 34274422,...etc), the review is comprehensive

Response: We would like to thank the reviewer for recognizing the comprehensive value of our manuscript, as indeed we tried to include all the information regarding crayfish. The ideas previously published were focused either on shrimp aquaculture (PMID: 33971259) or on probiotics administration in fish and selfish (PMID: 34274422), containing only a few refers to crayfish. It should be also noted that these references were already included in the manuscript and now further discussed in the revised version (references 33 and 143 in the revised manuscript). Furthermore, here we provide all the information regarding the administration not only of probiotics but also for prebiotics and synbiotics particularly in crayfish aquaculture and all the factors affecting crayfish microbiota.

Major points:

1- The authors need to highlight the new directions in this review that not published previously (i.e what are new data in this review that differs from previous published reviews discussing the same topic)

2-The authors mentioned in the Data Availability Statement: No new data were created or analyzed in this study. However, I see in the review data generated in Fig 1a, FIg 1b, Fig 2a, Fig 2b, and Figure. What are the source of these data used in these figures,

Response:

1- As clarified in both the Introduction and the conclusions, the new directions of the present study were to gather all the information available regarding the administration of probiotics, prebiotics and symbiotics in crayfish aquaculture, since no such review existed so far particularly for crayfish. In addition, our study summarizes all the factors that contribute to the alterations of the crayfish microbiome abundance and composition as well as the the microbiome baselines levels. All the above represent the first step towards the better understanding of the complex microbiome system in order to draw conclusions about the development of appropriate supplements that can be administered to solve the problems that may arise in the crayfish aquaculture industry.

2- Indeed, no new data were created in this study. In Figures 1-3 data from the literature were collected and analyzed mentioned in the text were collected and analyzed. Additionally, according to reviewers’ comment, in each diagram, the associated sources (references) were provided.

Reviewer 3 Report

Comments to authors:

-The current review is interesting; however, the authors should address the following comments to improve the quality of the manuscript:

Abstract:

-  The abstract must illustrate the urgent need for pro, pre as substitutes for antibiotics to cope the problem of antimicrobial resistance

- Line 30: please provide the full meaning of pros and cons

- Please amplify the conclusions of the previous findings and then give information about possible plan

Introduction:

- A classic odd introduction, with an inappropriate and poorly developed backstory. Authors should clarify the aim of this work, the limitations of previous studies, and new points to be addressed.

- Since crayfish farming is so low profile, the authors have to explain why they focus on this kind of production.

- Lines 88- 90: prefer to revise as: The huge demand for crayfish in the international market coupled with growing concerns about overfishing and the degradation of their natural habitats has helped this sector to grow in prominence

- Line 93: you take about facts and reports, so please provide a reference (which report provided by FAO organization?)

- Line 107: ratios or rations? Dietary supplement is not innovating technology; it is feeding regimen or management protocol.

- Line 108: symbiotics or synbiotics?

- Lines 109-111: please remove (redundant)

- Lines 115-117: Does not make sense. It was repeated several times above. Please remove

- The authors should mention the emergence of MDR isolates in aquaculture sectors and their rising public health threats, give hints about the mechanism of drug resistance and the possible use of these additives in different aquatic sectors. They can follow these publications:

doi.org/10.3390/fishes8020105

doi.org/10.1007/978-3-030-98621-6

- Line 143: please add references

- Table 1: all concentrations should be revised (log no should be superscript)

- Figure 1: is it your study? If not, please provide their source, references, and copyright

- The same for all figures

- Line 339: modes or models?

- Figures 4 and 5: Should be presented in a better way. It has many information. Try to be brief and use abbreviations  

-The manuscript should be revised for English editing and grammar mistakes.

Author Response

The current review is interesting; however, the authors should address the following comments to improve the quality of the manuscript:

Response: We want to thank reviewer, for finding our manuscript interesting and for the constructive comments that helped us improve the quality of our manuscript

Abstract:

- The abstract must illustrate the urgent need for pro, pre as substitutes for antibiotics to cope the problem of antimicrobial resistance

Response: According to reviewer’s suggestion, some parts were added in the abstract highlighting the urgent need for antibiotic alternatives for eliminating increased antimicrobial resistance.

- Line 30: please provide the full meaning of pros and cons

Response: In accordance to the reviewer’s comment “pros and cons” were replaced by “advantages and disadvantages”

- Please amplify the conclusions of the previous findings and then give information about possible plan

Response: Following the reviewers comment we tried to amplify the conclusions of the reviewed studies while at the same time we provided suggestions for future research directions (please see section 5. Conclusions and future perspectives)

Introduction:

- A classic odd introduction, with an inappropriate and poorly developed backstory. Authors should clarify the aim of this work, the limitations of previous studies, and new points to be addressed.

Response: The section entitled “Crayfish aquaculture” was merged with the introduction in an effort to enhance the clarity of the backstory, as suggested by the reviewer. Several parts of the Introduction were deleted and focus was given towards crayfish aquaculture. Further, we tried to develop the backstory, as more information regarding the increased resistant bacteria, as well as the urgent need for better understanding of the factors affecting crayfish microbiome in order to develop the optimal supplements, were highlighted. Further, the importance of understanding the complex system of crayfish intestinal microbiome was explained.

- Since crayfish farming is so low profile, the authors have to explain why they focus on this kind of production.

Response: This is very good point. Although indeed, crayfish aquaculture is a lower profile sector in comparison to shrimp aquaculture, its potential to be both economically and environmentally beneficial is rather considerable. Further, crustaceans are considered one of the foods with the fastest worldwide growth rates. So problems regarding crayfish aquaculture, should be gathered, highlighted and discussed in order to provide solution for further development. Additionally, crayfish aquaculture apart from having lower carbon footprint in comparison with other fish aquaculture practices, also contributes to the development of regional economy. This part was added in the Introduction in the revised manuscript.

- Lines 88- 90: prefer to revise as: The huge demand for crayfish in the international market coupled with growing concerns about overfishing and the degradation of their natural habitats has helped this sector to grow in prominence

Response: The sentence was revised according to the reviewer’s suggestion

- Line 93: you take about facts and reports, so please provide a reference (which report provided by FAO organization?

Response: Additional references from FAO and EU were added according to the reviewer’s comment

- Line 107: ratios or rations? Dietary supplement is not innovating technology; it is feeding regimen or management protocol.

Response: This specific sentence was replaced with “Among the strategies applied in reared aquatic species, one of the most promising is the utilization of live microorganisms’ administrated by injection, feed or as water additives for controlling infectious disease.”, in accordance to the reviewer’s comment (Please see lines 361-364 in the revised manuscript)

- Line 108: symbiotics or synbiotics?

Response: Although we found both terms in the literature, we chose to keep ‘symbiotic’, which was corrected in the whole manuscript.

- Lines 109-111: please remove (redundant)

Response: Corrected according to reviewers’ comment (lines 109-111 were removed)

- Lines 115-117: Does not make sense. It was repeated several times above. Please remove

Response: According to reviewers’ comment, lines Lines 115-117 were removed.

- The authors should mention the emergence of MDR isolates in aquaculture sectors and their rising public health threats, give hints about the mechanism of drug resistance and the possible use of these additives in different aquatic sectors. They can follow these publications:

doi.org/10.3390/fishes8020105

doi.org/10.1007/978-3-030-98621-6

Response: According to reviewers’ suggestions, information from the two suggested publications (doi.org/10.3390/fishes8020105; doi.org/10.1007/978-3-030-98621-6) were added, as well as these references. Further, the rising problem regarding the antibiotic resistant bacteria threatening the public health was discussed together with the urgent need of finding alternative supplements for crayfish aquaculture.

- Line 143: please add references

Response: According to reviewers recommendation the following ref was added: “Rohani, M.F.; Islam, S.M.; Hossain, M.K.; Ferdous, Z.; Siddik, M.A.; Nuruzzaman, M.; Padeniya, U.; Brown, C.; Shahjahan, M. Probiotics, prebiotics and synbiotics improved the functionality of aquafeed: Upgrading growth, reproduction, immunity and disease resistance in fish. Fish Shellfish Immunol. 2022, 120, 569-589.”

- Table 1: all concentrations should be revised (log no should be superscript)

Response: According to reviewers comment all concentrations revised, with log no turned into superscript

- Figure 1: is it your study? If not, please provide their source, references, and copyright

- The same for all figures

Response: Indeed, all the data presented in Figures 1-3 did not occurre from the present study. Thus, according to reviewers’ comment, in each figure, the source (reference) associated with the data presented was added.

- Line 339: modes or models?

Response: modes corrected into models, according to the reviewer’s comment

- Figures 4 and 5: Should be presented in a better way. It has many information. Try to be brief and use abbreviations

Response: As the reviewer suggested, we tried to present the information in a better way, by changing all the information to abbreviated, wherever possible.

Round 2

Reviewer 2 Report

No further comments

Minor editing

Reviewer 3 Report

The authors addressed all points.

Minor editings still required